# Rabs in Signaling and Embryonic Development

**DOI:** 10.3390/ijms21031064

**Published:** 2020-02-05

**Authors:** Sonya Nassari, Tomas Del Olmo, Steve Jean

**Affiliations:** Faculté de Médecine et des Sciences de la Santé, Department of Immunology and Cell Biology, Université de Sherbrooke, 3201 Rue Jean Mignault, Sherbrooke, QC J1E 4K8, Canada; sonya.nassari@usherbrooke.ca (S.N.); tomas.del.olmo@USherbrooke.ca (T.D.O.)

**Keywords:** rab signaling, embryonic development, membrane trafficking, enzyme catalyzed proximity labeling approach

## Abstract

Rab GTPases play key roles in various cellular processes. They are essential, among other roles, to membrane trafficking and intracellular signaling events. Both trafficking and signaling events are crucial for proper embryonic development. Indeed, embryogenesis is a complex process in which cells respond to various signals and undergo dramatic changes in their shape, position, and function. Over the last few decades, cellular studies have highlighted the novel signaling roles played by Rab GTPases, while numerous studies have shed light on the important requirements of Rab proteins at various steps of embryonic development. In this review, we aimed to generate an overview of Rab contributions during animal embryogenesis. We first briefly summarize the involvement of Rabs in signaling events. We then extensively highlight the contribution of Rabs in shaping metazoan development and conclude with new approaches that will allow investigation of Rab functions in vivo.

Membrane trafficking regulates molecule and protein transport between intracellular compartments and the extracellular milieu. Defects in membrane trafficking events are linked to a plethora of human diseases [1]. One important class of trafficking regulators is the Rab GTPases [2], which form the largest family of small GTPases, consisting of about 70 members in humans [3]. Like other GTPases, Rabs cycle between an inactive form, bound to guanosine diphosphate (GDP), and an active form, bound to guanosine triphosphate (GTP) [4]. GDP/GTP cycling is regulated by guanine nucleotide exchange factors (GEFs) and GTPase activating proteins (GAPs) [4]. Once activated, Rab GTPases localize to specific cellular compartments (Figure 1), where they recruit effectors that mediate their functions [5]. Rabs have been implicated in numerous cellular processes, and their roles in trafficking have been extensively reviewed [2,3,5].

Here, we compiled data highlighting Rab GTPases’ involvement in metazoan embryogenesis. Although, numerous studies have detailed the phenotypic contributions of Rabs in embryonic development, a limited number have described their molecular roles in this context. For that reason, we first discuss specific examples of Rabs in developmentally relevant signaling events. This with the goal of offering a potential hypothesis on putative Rab molecular roles during embryogenesis. Then, we highlight the requirement of Rabs during the embryonic development of various metazoan model organisms. Finally, we describe novel available techniques that will bring new insights on Rabs’ roles in signaling and their molecular contribution during embryogenesis.

## 1. Direct Involvement of Rab GTPases in Signaling

Depending on their cellular localization, activated receptors can differently affect signaling pathways, resulting in distinctive cellular responses. Intrinsically, membrane trafficking plays a central role in signaling regulation. Multiple studies have demonstrated the influence of Rab GTPases in signaling events. We will highlight a few of these studies in this section and discuss how Rabs are involved in whole-cell homeostasis by integrating and regulating intra- and extra-cellular signals such as amino acids, growth factors and Ca^2+^.

### 1.1. Regulation of the mTORC1 Signaling Pathway in Response to Amino Acids

One of the main sensors of intracellular amino acid levels is mammalian target of rapamycin complex 1 (mTORC1) [6]. In an environment rich in amino acids, mTORC1 phosphorylates S6 kinase (S6K), 4E-binding protein 1 (4E-BP1), and Unc-51-like autophagy activating kinase 1 (ULK1), amongst others, promoting cell growth and protein synthesis and inhibiting autophagy [7]. The activity of mTORC1 is dependent on the recruitment of mTORC1 to lysosomes upon amino acid stimulation [8,9]. Interestingly, an RNAi screen performed in *Drosophila* S2 cells revealed that knockdown of *Rab1*, *Rab5*, or *Rab11* reduced the phosphorylation of S6K by mTORC1 [9]. In the same study, the overexpression of the active forms of Rab5, Rab7, Rab10, and Rab31 in HEK293 mammalian cells also led to decreased S6K phosphorylation [9]. Thus, Rab5 seems to have an ambivalent effect on mTORC1 activity, where too much or too little Rab5 activity impairs mTORC1 signaling. These effects were also observed in another study, where the overexpression of Rab5-S34N (dominant negative form) and Rab5-Q79L (active and GTP-bound form) in 293A cells resulted in the relocalization of mTORC1 into the cytosol or Rab5-positive vacuolar structures, respectively [10]. A decreased phosphatidylinositol-3-phosphate (PtdIns(3)P) level was observed in yeast with a deletion in the *vacuolar sorting-associated protein 9* (*Vps9*) gene (a Rab5 GEF); therefore, proper PtdIns(3)P dynamics were found to be essential for amino acid-mediated mTOR activation. Thus, by regulating PtdIns(3)P levels, Rab5 affects mTORC1 signaling [10]. A recent study further established that PtdIns(3)P recruits two effectors that position lysosomes at the cell periphery, contributing to mTORC1 activity [11].

Compared to the indirect regulation of mTORC1 by Rab5, Rab1A was demonstrated to interact directly with mTORC1 under amino acid stimulation; Rab1A is localized at the Golgi, allowing mTORC1 recruitment and activation at this compartment [12]. Significantly, the overexpression of Rab1A in colorectal cancer was associated with overactivation of the mTORC1 pathway and a poor prognosis [12]. Finally, a recent study also established a role for Rab7 in the regulation of mTORC1 signaling. The overexpression of Rab7 in HeLa cells led to lower levels of phosphorylated S6K and 4E-BP1 [13]. This decreased activity was caused by the invasion of ‘ragulator microdomains’ by Rab7, which disrupted recruitment and activation of mTORC1 at lysosomes [13]. Importantly, this effect of Rab7 on mTORC1 is regulated by the retromer, which recruits Tre-2/BUB2/cdc1 (TBC1) domain family member 5 (TBC1D5) to inhibit Rab7 at endolysosomes [13]. These roles of the retromer and Rab7 on mTORC1 signaling are conserved, and knockdown of *Vps29* or *Vps35*, two retromer subunits in *C. elegans*, decreases the phosphorylation of 4E-BP1 and increases longevity [13]. Thus, multiple Rabs are involved in the recruitment of mTORC1 to different organelles, directly affecting mTORC1 activity.

### 1.2. Regulation of mTORC2 by Rabs

The mTORC2 complex can be activated by various growth promoting stimuli [14]. The role of a Rab protein in regulating TORC2 was first demonstrated in yeast. Using genetic approaches, the deletion of *Ryh1* (*Rab6* ortholog in *S. pombe*) was found to affect TORC2 signaling [15]. The authors showed that GTP-bound Ryh1 interacted with TORC2, and this interaction stimulated TORC2 signaling [15]. They further showed that both Ryh1 and TORC2 were required for vacuolar integrity [15]. Interestingly, human Rab6 expression was sufficient to rescue the *Ryh1* deletion, implicating Rab6 as a mTORC2 regulator [15]. The early endosomal protein Rab5 was also recently shown to modulate TORC2 signaling. Again, through a genetic screen, it was observed that the deletion of *Muk1* (a Rab5 GEF) affected TORC2 signaling [16]. Interestingly, it was also demonstrated that GTP-bound Vps21 (Rab5) interacted with TORC2 and stimulated its activity [16]. Deletion of all yeast *Rab5* paralogues (*Vps21*, *Ypt52*, and *Ypt53*) did not impair TORC2 localization to the plasma membrane, suggesting a direct role for Rab5 in stimulating TORC2 activity [16].

A direct connection between mTORC2 and Rab GTPases in mammalian cells remains to be identified. However, a genetic screen identified a few Rabs (*Rab1B*, *Rab35*, *Rab39A*, and *RabL3*) required for mTORC2 activity upon growth factor stimulation in HeLa cells [17]. Further characterization of Rab35 revealed an indirect effect of Rab35 on platelet-derived growth factor receptor α (PDGFRα) signaling and the activation of class I phosphoinositide 3-kinase (PI3K), resulting in mTORC2 activation [17]. Interestingly, activating mutations in *Rab35* were observed in various tumors and were sufficient to drive transformation [17]. With respect to oncogenic growth, Rab2A was recently demonstrated to directly bind to Erk1/2 and promote Erk1/2 signaling [18]. This interaction also enhanced the oncogenic growth of breast cancer cells [18].

### 1.3. Integration of Intracellular Ca^2+^ Concentration by Rabs

Modulation of mTORC activity by Rabs does not represent the only general signal detected and integrated by Rabs. Some atypical Rabs, including Rab44, Rab45, and Rab46 (CRACR2A), have a Ca^2+^-binding CLR recognition/interaction amino acid consensus (CRAC) domain. Rab44, localized to the Golgi and lysosomes, regulates lysosomal Ca^2+^. Indeed, the knockdown of *Rab44* induced a decrease in lysosomal pH and release of Ca^2+^ from the lysosomes. This increased cytosolic Ca^2+^ level activated nuclear factor of activated T cells 1 (NFATc1) and promoted osteoblast differentiation [19]. Rab46, another Rab with a CRAC domain, plays a role in the integration of Ca^2+^ signals [20]. During histamine stimulation of endothelial cells, Rab46 induced retrograde trafficking of a subpopulation of Weibel–Palade bodies (WPBs) along the microtubules to the microtubule organizing center (MTOC). This retrograde transport of WPBs is Ca^2+^-independent. However, the dispersion of WPBs accumulated at the MTOC required Ca^2+^ capture by Rab46, leading to Rab46 detachment from the microtubules [20].

### 1.4. Rab Regulation by Post-Translational Modifications

The connection between signaling and Rabs is not unidirectional. In view of their major roles in cellular homeostasis, Rabs are tightly regulated. In addition to GDP/GTP cycling, Rabs are subjected to numerous post-translational modifications (PTMs) that directly impact their activities.

One well-described PTM is Rab phosphorylation. Leucine-rich repeat kinase 2 (LRRK2), initially identified as being involved in Parkinson’s disease, influences a wide range of signaling pathways [21]. Interestingly, one of the preferential targets of LRRK2 is the Rab GTPase family [22]. Indeed, LRRK2 is involved in the phosphorylation of a threonine localized in the switch II domain of Rab1, Rab8, and Rab10, among other Rabs [23,24,25,26]. Parkinson pathogenic mutations in *Lrrk2* increased Rab phosphorylation, which decreased Rab binding to regulatory proteins like guanine nucleotide dissociation inhibitors (GDIs) [22]. LRRK2 is not the only kinase that catalyzes the phosphorylation of Rabs. Indeed, it has been shown that Src phosphorylates Rab7 on tyrosine 183. Phosphorylated tyrosine 183 decreases the interaction between Rab7 and its effector, Rab-interacting lysosomal protein (RILP), preventing epidermal growth factor receptor (EGFR) degradation [26]. In addition, Rab7 dephosphorylation at tyrosine 183 and serine 72 by phosphatase and tensin homolog (PTEN) induces EGFR degradation [27]. Another interesting example of regulatory PTMs of Rabs is the ubiquitinylation of Rab11 by HECT domain and ankyrin repeat-containing E3 ubiquitin-protein ligase (HACE1) in β-2 adrenergic receptor (β2AR) recycling [28]. In this context, GDP-bound Rab11 interacted with β2AR, and ubiquitinylation of Rab11 on lysine 145 by HACE1 facilitated Rab11 activation [28]. Once activated, Rab11 then induced β2AR recycling to the plasma membrane. Rab7 is also ubiquitinylated and deubiquitinylation of Rab7 by ubiquitin specific peptidase 32 (USP32) is involved in recycling events [29]. Indeed, USP32 depletion sequesters Rab7 in abnormally enlarged endosomal structures, and the accumulation of ubiquitinylated Rab7 promotes interactions with the retromer and inhibits interactions with RILP [29]. The absence of USP32 induced retromer sequestration in late endosomes and inhibited recycling events [29]. Interestingly, the palmitoylation of Rab7 was also recently shown to affect the Rab7 interaction with the retromer [30]. Thus, PTMs of Rabs are closely linked to their regulation and play direct roles in the recruitment of effectors and cargo.

### 1.5. Conclusions and Perspectives on the Role of Rabs in Signaling

Considering these examples, Rabs and cell signaling are closely linked. Rabs are activated in response to various stimuli and allow for the recruitment of effectors to specific compartments, modulating receptor transport and thus signaling. Rabs can also directly modulate kinase activity. In addition, signaling networks can induce PTMs that result in Rab activation or the regulation of Rab interactions with effectors. Thus, there is a complex link between signaling and membrane trafficking, allowing for the fine-tuning of cellular and organismal homeostasis. Coherent with this idea, many studies have highlighted requirements for Rab GTPases during metazoan embryogenesis (Table 1). Although their molecular roles during development are not fully established, the knowledge gained on Rabs’ contributions to signaling events will serve to elucidate their roles in development.

## 2. Rab GTPase Requirements in Embryonic Development

Most of the functional data on Rab GTPases—their intracellular localization, the effectors they bind, and their functions—derive from in vitro cell culture studies. Embryonic development is a highly complex process, requiring tightly coordinated cellular and morphological changes, which rely on membrane trafficking to properly address receptors, transporters and adhesion molecules. Using various model organisms, studies have shed light on the important roles of Rab proteins during development in vivo (Table 1). Significantly, although Rabs are broadly expressed during embryogenesis, they often display restricted or distinct expression patterns, which are usually conserved among species [31,32,33]. In this section, we aim to summarize the various requirements of Rabs during embryonic development.

### 2.1. Rab2

*Rab2* is highly expressed in the nervous system of both *C. elegans* [33] and *Drosophila melanogaster* [32,34]. In both species, loss of *Rab2* dramatically affects neuromuscular junctions (NMJs), which represent the contact site between a neuronal synapse and a muscle fiber [33,35]. In *C. elegans*, the *Rab2* ortholog (*UNC-108*) is crucial in establishing proper NMJ signaling [33]; however, *Rab2* mutants are also viable. Indeed, *Rab2* mutants display locomotion abnormalities, caused by improper formation of dense core vesicles at the Golgi and their aberrant fusion with late endosomal compartments [33].

In *Drosophila*, *Rab2* is crucial for survival, as *Rab2* mutants die at larval stages [36]. In larvae, Rab2 localized at NMJs in synaptic boutons, the terminal structures of axons, and muscles. Importantly, the ubiquitous knockdown of *Rab2* affects glutamate receptor levels at NMJs, suggesting a role for Rab2 in the regulation of glutamate receptor clustering, a key event in NMJ development [37]. In addition to its contribution in neurogenesis, a recent genetic screen revealed a function for *Rab2* in larval abdominal muscle remodeling during *Drosophila* metamorphosis [37]. Fujita et al. highlighted the contribution of autophagy in T-tubule disassembly and reassembly and showed that Rab2, with Rab7, are required for this process, increasing the clustering of autophagosomes with lysosomes [37].

Finally, mice have two *Rab2* paralogs: *Rab2a* and *Rab2b*. The complete loss of *Rab2a* is lethal [38]. Interestingly, heterozygous *Rab2a* mutants display nervous system defects [38].

### 2.2. Rab3

Neuronal expression of *Rab3* is conserved between *C. elegans* [39], mice [40], and *Drosophila* [34]. In *C. elegans*, Rab3 was specifically detected in synaptic vesicles. *Rab3* mutants are viable, although they showed mild behavioral defects with reduced locomotion and abnormal pharyngeal pumping and defecation [39]. In *Rab3*-mutant animals, the number of terminal synaptic vesicles in the neurons was dramatically reduced; however, neurotransmitter release was not affected [39]. These data suggest that *Rab3* contributes to synaptic organization and functions as a regulator of synaptic vesicle tethering or docking at the synapse excretory site [39].

Mammals have four different *Rab3* paralogs—*Rab3a*, *Rab3b*, *Rab3c*, and *Rab3d*—all of which have distinct, although slightly overlapping and mostly neuronal, expression patterns [41]. This is consistent with the redundant functions of mammalian *Rab3* paralogs. Indeed, knockout (KO) mice for two or three of the four *Rab3* genes are viable [42], while mice lacking all four *Rab3* genes die soon after birth. In mice, similarly to *C. elegans*, *Rab3* is required for vesicle recruitment at synaptic sites, despite not being crucial for synaptic vesicle exocytosis. Quadruple *Rab3* KO mice had no obvious brain abnormality, but synaptic responses were diminished [42], reinforcing the idea of a conserved role for *Rab3* in synaptic function. Furthermore, newborn *Rab3* mutants display cyanosis and an irregular breathing pattern. Interestingly, newborn lethality could be partially bypassed in a high-oxygen atmosphere. This phenomenon is consistent with the impaired respiration observed in those mutants and highlights a role for *Rab3* in proper establishment of the respiratory system [42].

### 2.3. Rab5

Mammals have three *Rab5* paralogues; zebrafish have four [43], while flies and worms have only one [44].

In *Drosophila*, a *Rab5* mutant revealed an important role for this Rab in synaptic function, endosomal trafficking, and egg cellularization [45]. In larvae, *Rab5* is required for neurotransmitter release and synaptic vesicle recycling [45]. *Rab5* mutations are lethal in flies, which die at the larval stage. These mutant flies display weak locomotion defects. Interestingly, *Rab5* expression, specifically in neurons, is sufficient to rescue the *Rab5* mutant lethality, illustrating the importance of *Rab5* in neuronal functions [45].

In *Xenopus* embryos, Rab5 was detected in the developing retina and localized to growth cones, corresponding to the end of the elongating axon [46]. Loss of Rab5 using morpholinos delayed axon elongation [46], highlighting a conserved function for *Rab5* in neurons.

Multiple trafficking genes are maternally contributed, and the ability of *Rab5*-depleted flies to reach the larval stage is dependent on the maternal Rab5 protein. Hence, *Rab5* germ-line mutants, in which the maternal contribution of Rab5 is removed, show early embryonic lethality [45]. Moreover, during oogenesis, Rab5 is enriched close to the oocyte plasma membrane and contributes to yolk endocytosis, which is necessary for the proper development of the egg. Oocytes depleted of *Rab5* display defective yolk protein internalization [47]. Interestingly, it was recently shown in *C. elegans* embryos that Rab5 was enriched at cleavage furrows [48]. Depletion of *Rab5* in four- and six-cell embryos caused multinucleated blastomeres, revealing a role for *Rab5* in cytokinesis in *C. elegans* [48]. Notably, *Rab5* was shown to be important for cell boundary formation during fly cellularization [49].

In zebrafish, *Rab5* also contributes to embryo formation. As such, it mediates nodal signaling implicated in dorsal/ventral (DV) axis formation [43]. Loss of the *Rab5ab* ortholog in zebrafish is lethal [43], and the embryos do not form a dorsal organizer, which is required for the proper DV axis formation [43].

In mice, independent deletions of *Rab5a* and *Rab5b* are viable [38], while loss of *Rab5c* is lethal [38]. Heterozygous mutants for *Rab5c* show hematopoietic and metabolic defects [38].

### 2.4. Rab6

Two *Rab6* paralogues are present in *C. elegans—Rab6.1* and *Rab6.2* [50]. *Rab6.1* and *Rab6.2* are broadly expressed in the intestine, muscles, gonads, vulva, and neurons. Individual *Rab6.1* or *Rab6.2* mutants are viable, although *Rab6.2* mutants show a cuticle phenotype [51]. Inactivation of both *Rab6.1* and *Rab6.2* is lethal, highlighting a likely redundancy in their functions. Expression of *Rab6.2* is detected in the epidermis [51] and is required for proper structural integrity and permeability of the cuticle. *Rab6.2* KO increased the number of ruptured *C. elegans* under normal growth conditions due to a fragile cuticle [51]. Importantly, *Rab6* regulation and function are phylogenetically conserved because *Rab6.2* KO *C. elegans* is rescued by mammalian *Rab6a* [51].

In mammals, there are two genes for *Rab6* [52]—*Rab6a*, and *Rab6b*. *Rab6a* expression generates two distinct isoforms that differ in three amino acids due to alternative splicing. Mice lacking *Rab6a* die early in development, at embryonic day seven [52], highlighting an important and unelucidated role for *Rab6a* in early mouse embryogenesis.

In *Drosophila*, depletion of *Rab6* is associated with a closed rhabdomere phenotype [53].

### 2.5. Rab7

In humans, *Rab7* is ubiquitously expressed, with the highest expression in skeletal muscles; in contrast, in mice, *Rab7* is highly expressed in the liver, heart, kidneys, and specific subpopulations of neurons [54]. Importantly, missense mutations in *Rab7* are associated with Charcot-Marie-Tooth type 2B neuropathy, characterized by muscle weakness and wasting [54]. These data suggest a role for *Rab7* in establishing correct communication between muscles and neurons at NMJ during development.

KO of *Rab7* in mice is lethal. In such mutants, embryogenesis is dramatically affected, and developing embryos die around embryonic day seven to eight [55]. *Rab7* mutant embryos fail to complete gastrulation, which results in a disorganized mesoderm [55]. The *Rab7* loss strongly perturbs endolysosomal organization, affects signaling events required for the establishment of the embryo anterior–posterior axis [55].

In *Drosophila Rab7* null mutants are lethal at the pupal stage [56].

### 2.6. Rab8

In mammals, two genes encode *Rab8* paralogues—*Rab8a* and *Rab8b*. Deletion of either one does not impact viability [57,58], although *Rab8a* KO affects microvilli formation in the intestine [57], as well as body fat amount [38]. However, *Rab8a* and *Rab8b* double-KO mice die two weeks after birth [58]. This highlights the functional redundancy of *Rab8a* and *Rab8b* during embryogenesis. Mice lacking both *Rab8* genes display severe defects in intestinal epithelium morphology, with a complete atrophy of the enterocyte microvilli and microvillus inclusion bodies [58]. These observations suggest a role for *Rab8* in the formation of the intestinal epithelium. Interestingly, in vitro studies have shown a role for *Rab8* in cilia formation [59]. Surprisingly, double-KO mice had normal cilia. Because *Rab10* and *Rab13* are in the same subfamily as *Rab8*, the authors found that in a developmental context, *Rab10* could complement *Rab8* loss [58].

In *Drosophila* embryos, *Rab8* expression and localization are highly dynamic during development. Indeed, during embryo cellularization, Rab8 oscillates between punctate and membrane-associated domains [60]. Thereby, Rab8 participates in membrane delivery to the furrow during cellularization. Mutants for *Rab8* are lethal and die at the pupal stage [61]. Also, knockdown of *Rab8* or the expression of a Rab8 dominant negative form impairs gastrulation movements and further affects embryonic development.

### 2.7. Rab9

In *Drosophila*, Rab9 participates in the establishment of the tracheal system [62]. In this context, Rab9 regulates the endosomal recycling of the luminal protein serpentine, which is involved in chitin assembly in the trachea. As a consequence, the total loss of *Rab9* in fly embryos results in shortened tracheal tubes, and is lethal [62].

### 2.8. Rab10

Mouse embryos depleted of *Rab10* die at embryonic day 9.5. In these mutants, development of the three primary cell layers is stopped at embryonic day 7.5, and the embryos are much smaller compared with wild-type embryos [63]. Consistent with this observation, *Rab10* KO embryos are deficient in proliferation at this stage [63]. At day 8.5, no recognizable structures are present in KO embryos [63]. *Rab10* is thus crucial for the early steps of embryogenesis, although the exact mechanism controlled by Rab10 during embryogenesis remains to be defined.

Surprisingly, in *Drosophila*, a null *Rab10* allele is viable [64].

### 2.9. Rab11

In mouse embryos, *Rab11a* and *Rab11b* display general but distinct expression patterns, with *Rab11a* detected in the heart and *Rab11b* detected in the forelimb of E10.5 and E11.5 embryos, respectively [65]. *Rab11a* is involved in the secretion of metalloproteinases during the implantation stage [66]. The global loss of *Rab11a* in mammals is not viable, and embryos die in utero around the implantation stage [66]. *Rab11a* mutants are unable to implant in the uterus, likely because extracellular matrix degradation is impaired. Mutants for *Rab11b* are viable, with no obvious phenotypes.

In *C. elegans*, *Rab11* is detected in oocytes, and depletion of *Rab11* affects the exocytosis of cortical granules [67]. In addition, *Rab11* knockdown in *C. elegans* embryos leads to polarity defects related to the function of *Rab11* in spindle alignment [68]. A complete loss of *Rab11* is lethal in *C. elegans* [67].

In *Drosophila*, *Rab11* is highly expressed in the developing central nervous system of embryos, with enriched expression during axon formation [69]. Total depletion of *Rab11* in fly embryos is lethal [70]. The partial depletion of *Rab11* in neurons leads to morphogenesis defects, characterized by misrouted axons [69], highlighting the contribution of *Rab11* in proper morphogenesis of the nervous system.

Similarly, zebrafish have three *Rab11* paralogs that are strongly expressed in the embryo brain, heart, kidney, ovary, skin, and muscles [71].

Finally, in *Xenopus*, *Rab11* is detected at the animal pole of blastula-stage embryos, with expression restricted to the neural folds and notochord at the neurula stage [72]. *Rab11* is important for proper embryo elongation by participating in planar cell polarity [73]. It is also required for proper establishment of left-right patterning. This role of *Rab11* might be mediated by its contribution to ductin recycling to the cell surface, which is required for proper lateral formation [72]. Expression of a Rab11 dominant negative form in *Xenopus* embryos dramatically impairs left-right patterning.

### 2.10. Rab13

In zebrafish embryos, *Rab13* contributes to trunk vasculature morphogenesis. Indeed, upon morpholino-mediated *Rab13* knockdown, intersegmental vessel sprouting is arrested. This phenotype is likely due to the involvement of *Rab13* in regulating cell migration during angiogenesis [74].

In mice, surprisingly, *Rab13* is not crucial for embryogenesis and *Rab13* KO are viable with no significant phenotypes [38].

### 2.11. Rab14

In mammalian embryonic stem cells, *Rab14* regulates the transport of fibroblast growth factor receptor 2 (FGFR2) to the plasma membrane through its interaction with kinesin family member 16B (KIF16B), which is crucial for FGF signaling during embryogenesis [75]. Significantly, the expression of a Rab14 dominant negative mutant in embryonic stem cells delayed cystic embryonic body formation [75].

### 2.12. Rab17

In mice, *Rab17* expression is developmentally regulated in the hippocampus, and its localization in neurons is restricted to specific regions (dendritic growth cones, spines, and shafts) [76]. In line with its localization, *Rab17* is necessary for correct post-synaptic and dendritic development. The depletion of *Rab17* using small hairpin RNAs (shRNAs) in hippocampal neurons induced reductions in the dendrite branch number and dendritic spines, while the overexpression of *Rab17* promoted dendrite formation [76].

### 2.13. Rab18

In mammals, *Rab18* is involved in brain, eye, and central nervous system function [77]. *Rab18* mutant mice are viable and fertile [78,79]; however, the mutants display phenotypic defects reminiscent of the clinical phenotypes observed in Warburg micro syndrome (WARBM) patients [77,78]. This pathology defines a developmental disorder characterized by brain, eye, and endocrine defects. At embryonic day 12.5, *Rab18* mutant mice have delayed lens development [79]. When the lenses are formed, vacuoles are observed at the lens periphery, characteristic of cataract development. *Rab18* KO mice also display impaired NMJs with neurofilament accumulation [79].

In *C. elegans*, *Rab18* mutants are viable and show decreased lipid content [80], suggesting a putative involvement of Rab18 in lipid homeostasis.

### 2.14. Rab21

Unpublished data from our laboratory show that the complete loss of *Rab21* in *Drosophila* is not critical for proper embryogenesis; *Rab21* mutants are viable. In mice, however, the complete loss of *Rab21* is lethal [38], but the underlying reason for the lethality remains to be defined.

### 2.15. Rab23

In mammals, *Rab23* is expressed during embryonic development and enriched in the brain [81]. *Rab23* is necessary for nodal expression in the lateral plate mesoderm and contributes to the establishment of correct left–right patterning [82]; this role is conserved in zebrafish. Moreover, the loss of *Rab23* in mice is lethal [82], partly because of its role in chondrocyte development [81]. Significantly, in mice and humans, *Rab23* mutations lead to defects in cartilage and bone development.

Interestingly, *Rab23* mutations, which negatively affect hedgehog (Hh) signaling, are found in Carpenter syndrome, characterized by craniofacial, polydactyly, and soft-tissue syndactyly defects. This suggests an important role for *Rab23* in craniofacial and limb morphogenesis [81].

### 2.16. Rab25

In mouse embryos, *Rab25* expression is detected at stage E14.5 in several anatomical regions, e.g., the stomach, pancreas, midgut, epidermis, and pharynx epithelium [65]. Rab25 is an atypical Rab because it is believed to be constitutively active [83]. *Rab25*-deficient mice are viable, with no obvious phenotypes [84]. However, *Rab25* mutants were recently shown to display defective skin homeostasis [85]. Such results suggest a role for *Rab25* in regulating epidermal formation [85].

### 2.17. Rab26

In mice, complete loss of *Rab26* is non-lethal [86], although the lungs are dramatically affected compared with wild-type mice following LPS stimulation, based on histological examination [86]. In this context, *Rab26* is required for adherens junction maintenance, through its regulation of Src and ATG16L1.

### 2.18. Rab27

Two *Rab27* paralogs exist in mammals—*Rab27a* and *Rab27b. Rab27a* is highly expressed in the lung, gastrointestinal tract, spleen, skin, and eye in mammals, with lower expression in the liver, heart, testis, and kidney [87]. Both Rabs have been extensively studied for their importance in exosome secretion [88]. In mouse embryos, *Rab27* is highly expressed in oocytes, but its expression decreases in fertilized oocytes and developing embryos, until expression is stopped at the blastocyst stage [87]. *Rab27a* KO mice are viable and fertile [87]. However, mutations in *Rab27a* in humans are associated with Griscelli syndrome [89], characterized by skin and hair pigmentary loss, with melanosome accumulation in melanocytes. *Rab27a* and *Rab27b* double-KO mice are viable and display low-grade chronic inflammation due to exosome secretion deficiency [90].

### 2.19. Rab28

In *C. elegans*, Rab28 localizes to the ciliary structures associated with sensory neurons [91]. Rab28 showed activity-dependent intraflagellar transport (IFT) and deregulation of *Rab28*-induced defects in the formation and function of sensory pores [91], suggesting a role for *Rab28* in the formation of sensory neurons. Importantly, *Rab28* required the Bardet–Biedl syndrome-related complex for ciliary localization. In mice, the loss of *Rab28* did not impact viability and fertility but showed progressive cone degeneration, reminiscent of the phenotype observed in humans with mutated *Rab28* [92].

### 2.20. Rab32

In *C. elegans*, *Drosophila*, and vertebrates, *Rab32* has been associated with the formation of lysosome-related organelles (LROs) [93]. Mutations in *Rab32* affect eye color in flies, while mutations in *GLO-1* (a *Rab32* ortholog in *C. elegans*) affect gut granule biogenesis in worm embryos [93].

In mice, *Rab32* loss is subviable and mutant males show increased bone mineralization, while both males and females have decreased circulating alanine and aspartate transaminase levels [38].

### 2.21. Rab34

In mammals, Rab34 localizes to the cilia [94], and *Rab34* mutant mice are non-viable. Mutant embryos display polydactyly, cleft-lip, and palate defects [94]. Ciliogenesis is strongly impacted in *Rab34* mutants [94], with an observable decrease in cilium density, rendering *Rab34* mutant cells unable to respond to Hh signaling. Mutant embryos survive until late embryogenesis (E18.5) and do not exhibit morphological defects in the heart, lungs, or liver, indicating that Rab34-mediated Hh signaling is restricted to specific tissues or compensated by other Rabs involved in the regulation of Hh signaling [94]. *Rab34* mutants also display craniofacial malformation, hemorrhage, edema, and exencephaly [38]. Interestingly, a broad variability in *Rab34*-associated phenotypes is observed at E18.5 [38].

### 2.22. Rab35

In *Drosophila* embryos, Rab35 is detected at constricting surfaces, and loss of *Rab35* results in mesoderm invagination failure [95]. This is caused by the mislocalization and accumulation of myosin II in the cytoplasm. Furthermore, Rab35 is required for collective cell migration during salivary gland development [96].

In *C. elegans* embryos, Rab35 was detected in the pseudopods of engulfing cells during apoptotic cell clearance [97]. During this process, Rab35 is also localized in forming phagosomes and slowly decreases during engulfment. This localization pattern suggests multiple contributions for Rab35 at different steps of apoptotic cell clearance [97]. Not surprisingly, *Rab35* knockdown negatively affects the clearance of apoptotic cells [97].

In mice, loss of *Rab35* is lethal, and *Rab35* heterozygous mutants display defects in lens formation and develop cataracts [38].

### 2.23. Conclusion on the Roles of Rabs in Development

From the studies described above, Rab GTPases play important and non-redundant functions during embryonic development (Table 1). Furthermore, these studies exemplify the panoply of pathways affected by Rabs. Although the cited studies have been instrumental in providing an understanding of the physiological functions of specific Rabs, they in most cases rely on KOs or RNAi-mediated knockdowns. Compensation from other genes or Rabs could preclude identification of specific phenotypes in a developmental context [98]. Hence, to better understand the responses of Rabs to stimuli in cells and organisms, it is essential to develop new experimental approaches, capitalizing on proteomics and genetic tools [99]. In the last section, we will briefly discuss new approaches that will help to assess new Rab functions and thus to better characterize their molecular roles and the signaling pathways they affect during embryogenesis.

## 3. Proximity Labeling Approaches to Map Rab Neighbors and Interactors

The isolation of Rab GTPases for interactor studies by conventional immunoprecipitation approaches has been hampered by several technical issues. First, Rabs are associated with membranes once activated, which complicates their extraction [3]. Secondly, they often show basal GTPase activity, which leads to GTP hydrolysis during purification and results in the loss or weak recovery of potential interactors [101]. Finally, GEF/Rab or GAP/Rab interactions are transient, and the purification of these complexes is difficult [102]. Hence, constitutively GDP- or GTP-bound mutants are often used to identify Rab GEFs or GAPs. Although this approach has proved effective on various occasions, these mutants often do not localize to the appropriate cellular compartments [101]. Given all these difficulties, a limited number of studies have used conventional immunoprecipitation coupled to mass spectrometry to define Rab interactors.

The most effective and classical approach used to identify Rab effectors are pull-down experiments [103]. This type of approach has allowed the identification of most Rab effectors to date and is highly efficient for identifying direct effectors [104,105]. Unfortunately, pull-down experiments do not preserve the signaling state of the cells, and stimuli-dependent effectors cannot be purified because the cellular context is lost through the in vitro purification. Nonetheless, high-throughput studies on a large number of Rabs have identified important Rab effectors [104,105]. Likewise, yeast two-hybrid (Y2H) studies have screened binary interactions between Rabs and binding proteins and identified important regulators or effectors [106]. However, like in the pull-down studies, Y2H does not allow for the analysis of dynamic interactors in the cellular context.

To palliate the drawbacks of these approaches, we have recently demonstrated the usefulness of ascorbate peroxidase 2 (APEX2) proximity labeling of Rab neighbors (Figure 2A) [101]. APEX2 is a peroxidase that, upon brief H_2_O_2_ stimulation in the presence of biotin-phenol, allows the covalent biotinylation of proximal proteins [107]. When APEX2 is fused to a Rab, H_2_O_2_ stimulation delivers a snapshot of Rab neighboring proteins at a given time. By mapping early endosomal Rabs, we showed that APEX2 enabled biotinylation of a large array of proteins and allowed the identification of protein complexes, GEFs, GAPs, and effectors [101]. One important limitation of this approach, however, is the large number of identified proteins [101], which makes hit prioritization difficult. To palliate this drawback, we hope that ‘mapping’ most Rabs will help in identifying specific Rab neighbors and thus facilitate hit prioritization, something which is currently undergoing in the laboratory. Another advantage of APEX2, given the rapid labeling, is that it can be used to interrogate Rab environments under various stimuli. Preliminary data from our laboratory have shown the feasibility of this approach using quantitative mass spectrometry approaches such as stable isotope labeling by amino acids in cell culture (SILAC).

Recently, another approach was reported, using targeting of BirA (a promiscuous *Escherichia coli* biotin ligase)-fused Rabs to mitochondria (Figure 2B), followed by identification of biotinylated proteins by mass spectrometry [104]. In comparison to APEX2, this novel approach allowed the clear identification of GEFs, GAPs, and effectors and proved efficient for an array of small GTPases [104]. However, protein complexes known to interact with specific Rabs were not detected (i.e., Rab7 and the retromer). Also, its use to identify stimuli-dependent effectors might be limited, given that Rabs are ectopically expressed [104]. Nonetheless, the lower number of neighboring proteins identified by this approach simplifies hit prioritization over the APEX2 method.

These two proximity-labeling techniques are successful when applied to cell culture studies but difficult to use in vivo. Recently, two highly efficient variants of BirA were developed, miniTurbo and TurboID (Figure 2C), and both were shown to be efficient in vivo in *C. elegans* and *Drosophila* [105]. Basically, transgenic animals expressing a biotin ligase fused to a protein of interest are fed biotin, which allows in vivo biotinylation and characterization of neighboring proteins (Figure 2C). Hence, using these novel enzymes fused to Rab GTPases will allow mapping of cell type specific protein neighbors in vivo. Moreover, performing in vivo proximity-labeling in the context of various types of mutant could allow better definition of Rab regulation in tissues. As an example, one could knockdown a specific Rab GEF and assess neighboring proteins of a miniTurbo tagged Rab in a tissue to understand how deletion of its GEF affect effector recruitment.

## 4. Conclusions and Perspectives

Altogether, this review illustrates the important roles played by Rabs in signaling events and during metazoan development. Numerous Rabs are crucial for embryonic development to proceed correctly, and previous studies have demonstrated their pleiotropic contributions in this process. However, their specific molecular roles in vivo need to be further investigated, but these might be more easily sought through the novel proteomic approaches described above. Nonetheless, classical loss-of-function or gain-of-function approaches will still be required to define the functional relationships between Rabs and newly identified interactors/neighbors. Given the advent of clustered regularly interspaced short palindromic repeats (CRISPR)/CRISPR-associated protein 9 (Cas9) to generate null or knock-in alleles, it will be significantly easier to perturb Rab function and analyze its impact on cell function. Lower model organisms like *C. elegans* or *Drosophila* are particularly suited for such studies, given the plethora of genetic tools and the relative simplicity of genome manipulation in these organisms [108]. Moreover, these model systems have proved highly reliable to ask cell biology-relevant questions in the past. Given new proteomic approaches to define Rab interactors and the numerous CRISPR/Cas9 tools available to modify genomes, we believe that exciting discoveries will be made on the roles of Rab GTPases in coming years.

## Figures and Tables

**Figure 1 ijms-21-01064-f001:**
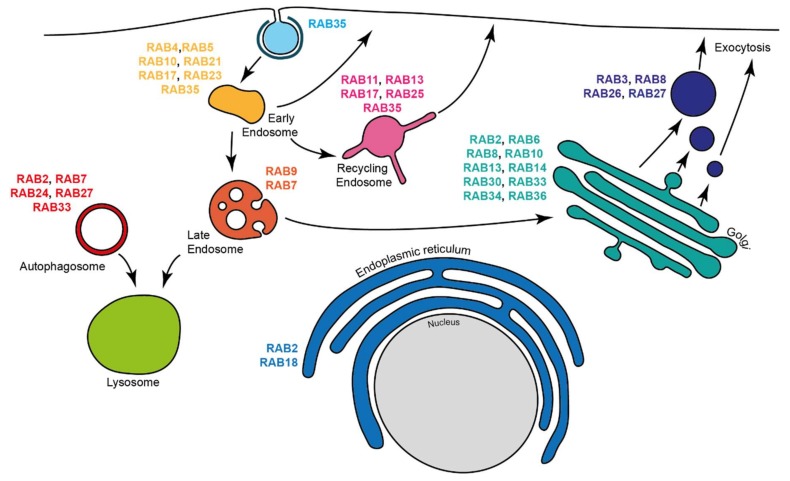
Rabs subcellular localization. Scheme illustrating the major cellular localizations of various Rabs.

**Figure 2 ijms-21-01064-f002:**
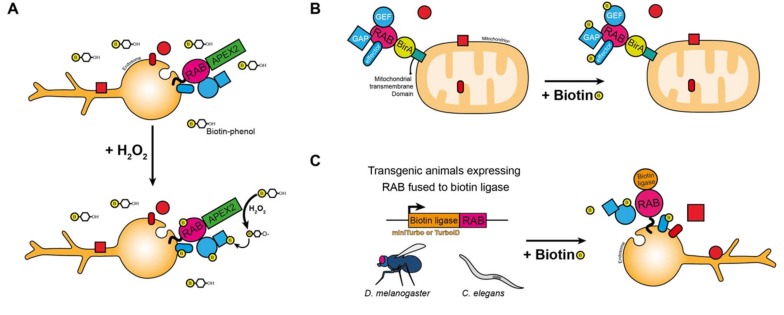
Proximity labeling approaches to characterize Rabs neighboring proteins in vitro and in vivo. (**A**–**C**) Scheme illustrating the different proximity labeling approaches used in vitro (**A**,**B**) and in vivo (**C**). (**A**) APEX2, (**B**) BirA and (**C**) miniTurbo or TurboID.

**Table 1 ijms-21-01064-t001:** Rab mutant-associated phenotypes in various metazoan model organisms.

	*M. musculus*		*D. melanogaster*	*C. elegans*	
***Rab2***	*Rab2a*	**Lethal**. Heterozygotes display neuronal and metabolic defects [38].	**Lethal** at L2/L3 larval stage [36].	**Viable**, with locomotion defects [33].	
	*Rab2b*	No data			
***Rab3***	*Rab3a*	**Viable** [42].	No data	**Viable**, with mild behavioral abnormalities [39].	
	*Rab3b*	**Viable** [42].			
	*Rab3c*	**Viable** [38,42].			
	*Rab3d*	**Viable** [42].			
	*Rab3a, b, c, d*	**Newborn Lethal**. Display cyanosis, with irregular breathing. Synaptic responses are diminished [42].			
***Rab5***	*Rab5a*	**Viable**. Nervous and hematopoietic system defects [38].	**Lethal** at larval stage. Mutants display weak locomotion defects [45].	**Lethal** [48].	
	*Rab5b*	**Viable**. Nervous systems impacted (hyperactivity) [38].			
	*Rab5c*	**Lethal**. Heterozygotes display hematopoietic and metabolic defects [38].			
***Rab6***	*Rab6a*	**Lethal** at embryonic day 7 [52].	**Viable** [53].	*Rab6.1*	**Viable** [51].
				*Rab6.2*	**Viable**, with fragile cuticle [51].
	*Rab6b*	**Viable**. Abnormal behavior with a decrease in total body fat amount [38].		*Rab6.1* and *Rab6.2*	**Lethal** [51].
***Rab7***	**Lethal** around embryonic day 7 to 8 [55].		**Lethal** at pupal stage [56].	No data	
***Rab8***	*Rab8a*	**Viable** [57,58]. Microvilli formation in the intestine is impacted [57] and total body fat amount is increased [57,58].	**Pharate lethality** [61].	No data	
	*Rab8b*	**Viable** [57,58].			
	*Rab8a, b*	**Lethal** two weeks after birth. Intestinal morphology is impacted [58].			
***Rab9***	*Rab9a*	No data	**Lethal** with impaired tracheal tubes establishment [62].	No data	
	*Rab9b*	No data			
***Rab10***	**Lethal** at embryonic day 9.5 [63]. Heterozygotes display nervous and cardiovascular impairments, and abnormal retina morphology [38].		**Viable** [64].	No data	
***Rab11***	*Rab11a*	Lethal at implantation stage [66]. Heterozygotes display growth retardation with eye morphology defects [38].	**Lethal** [70].	**Lethal** [67].	
	*Rab11b*	**Viable** [38].			
***Rab13***	**Viable** [38].		No data	No data	
***Rab14***	No data		No data	No data	
***Rab17***	**Viable** [38].		No data	No data	
***Rab18***	**Viable**. Mutants display lens development delays and NMJs defects [80].		No data	**Viable** [80].	
***Rab21***	**Lethal** [38].		**Viable** (S. Jean’s laboratory, unpublished data).	No data	
***Rab23***	**Lethal** [38,82]. Mutants display defects in cartilage and bone development [81].		No data	No data	
***Rab25***	**Viable** [38,84] with skin homeostasis defects [85].		No data	No data	
***Rab26***	**Viable** [86].		No data	No data	
***Rab27***	*Rab27a*	**Viable** [87].	No data	No data	
	*Rab27b*	**Viable**, with abnormal sleep behaviors [38].			
	*Rab27a, b*	Viable, with low grade global chronic inflammation [90].			
***Rab28***	**Viable**. Mutants display progressive cone degeneration [92].		No data	**Viable** [91].	
***Rab32***	**Subviable**. Males show increased bone mineralization and both sexes have mild metabolic defects [38].		**Subviable** [100].	No data	
***Rab34***	**Lethal** at embryonic day 18.5 [94]. Embryos show craniofacial malformations and limb morphology defects [38,94].		No data	No data	
***Rab35***	**Lethal**. Heterozygous embryos display abnormal lens morphology and cataracts [38].		No data	No data	

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
