# Peer review of "Rabs in Signaling and Embryonic Development"

_ijms, 2020, doi:10.3390/ijms21031064_

Round 1

Reviewer 1 Report

This is an interesting review on a complex topic, given the large number of Rab proteins and the potential for functional overlap among them. The authors have made a great effort to compile and summarize information about the roles of Rabs in embryonic development, so this review should be useful to those wishing to gain an overview of the contribution of Rabs to that particular aspect. While the review is well written and logically organized, my immediate impression after reading it is that the authors are dealing with three main sections (signalling, embryonic development and Rab interactome) that seem to me quite independent in the sense that neither section appears to support or justify the other. My suggestion would be either find a clear link between sections or modify the title to broaden its scope (something like “Rabs in signalling and embryonic development”; just an idea). The last part (the interactome) touches an interesting and timely aspect and would benefit from some more elaboration and the inclusion of a figure to illustrate the various techniques.

Another major aspect concerns the Table. This is a key resource but looks incomplete and seems to contain some errors. More specifically:

Include references in the appropriate cells. Only one is mentioned in D. melanogaster Rab2 (with the wrong formatting). The table should include more phenotypical information; not necessarily extensive, but clearly more. For example, for Rab23, mention defects in cartilage and bone development; for Rab25, defective skin homeostasis; for Rab28, progressive cone degeneration. Whether the phenotype is given as viable or lethal, this additional information is important to highlight for what process each Rab is important. Why not including data on silencing/partial depletion mutants or mutants expressing constitutively active or dominant negative Rabs? This will likely apply to fish and lower organism models and including that information should be achievable, judging from the look at how many cells in the table are marked “No data”. Consider removing zebrafish from the table for simplicity; there is only data for two Rabs. Similar situation with Xenopus, which is not in the table. Unless adding silencing data fills more cells. The information in the text and the table does not always match. For example, for Rab6A the table says No data, the text says mice lacking Rab6a die early in development. Same for Rab7, Rab8, Rab26, Rab27. Rab10 is given as viable in the table but dies at E9.5; what is the criterium for viability? If a mutation proves lethal, why not giving information throughout the table about the embryonic development day when known? Rab35 appears as lethal in the table but there is no mention in the text. The table mentions Rab7a and Rab7b, the text doesn’t mention these two genes. I’m not sure the references given in the text for the Rab7 KO (#66 and 67) are the correct ones. The authors should carefully verify the data in the table and match it with the text; I have mentioned the errors I could spot but my list is not exhaustive. The table should be referenced before the Rabs are described in detail, not at the end of the section.

Other comments:

-I recommend verifying the referencing throughout, I don’t have the impression the literature appears correlatively in the text. For example there is a jump from #59 to #66,67 in the Rab7 section and the missing references appear later in the Rab8 section.

-A few acronyms need spelling out: TBC1 (line 70), HACE1 (line 152)

-Mouse gene nomenclature is Rab3a, Rab3b, etc (only first letter capitalized), like in the table; same for other Rabs with isoforms, therefore not Rab3A. Please correct in the text.

-Some mentions to Rabs seem to refer to the gene and therefore should go in italics: Rab21 (line 122), Rab2 (line 182). Also Drosophila in line 192.

-Lines 99-110, the 2+ of the calcium ion should go as superscript.; line 442, the 2’s of H2O2 as subscripts.

-Figure 1. Could you mirror “Nucleus” so that it can be read without having to rotate the page?

-line 188: Rab2 localizes at… (rather than is expressed, to avoid confusion with gene expression).

-lines 211-212, delete “purplish skin coloration”. There is no need to define scientific terms.

-If I’m correct, IJMS requests a list of abbreviations at the end of the paper.

Reviewer 2 Report

The manuscript entitled “Contribution of Rabs in shaping embryonic development” by S. Nasari et al. aims to review the literature on this subject.

General comments:

This review represents a very broad and very ambitious task. It is presented in three parts dealing with 1/ direct signaling of Rabs, 2/ Rabs’ involvement in embryogenesis and 3/ technical issues to determine Rabs interactors.

The three parts appear to be very independent and I found difficult to understand how signaling by the different Rabs (1st part) were indeed involved in embryogenesis (2nd part). The third part dealing with Rabs interactors, is mainly a technical prospective concerning signaling and should not separated from part 1. Rabs GTPases are essentially involved in cell membrane organization and thus of organelles as well as in membrane receptors recycling. It is not clear in the review, how these membrane movements and receptors internalization and recycling control cell fates during embryogenesis. Parts 1 and 2 should be merged in a way that permits to see the links between the three steps: Rabs signaling (upstream and downstream partners) membrane and organelles movements and   The review does not take the plants into consideration. The title should mention “…development in metazoa (or animals)” to acknowledge this.

Specific comments:

Line 36: which activated receptors? Membrane receptors I guess. Also GPCR ? In part 1. Explain the choice of the described pathways (mTORC1, mTORC2, Ca++, TLRs) relative to embryogenesis. Why describe inflammation pathway in the scope of embryogenesis ?   Do all Rabs strictly have the same orthologs in all animal species ? Paragraph 4 (conclusions and perspectives) only relates to part 3. The word “these” (line 468) is vague. The paragraph “Author contributions” can be removed in a Review.

Round 2

Reviewer 1 Report

The authors have address my comments satisfactorily and the new version is much improved.

Reviewer 2 Report

The authors have answered very satisfactorily to the question I raised.

I recommend that this review can be published in IJMS.

Since IJMS is a multidisciplinary journal, I keep recommending that "animal" or "metazoa" is mentioned in the title. It is up to the Editors to decide whether it is important or not.